# Cangrelor Induces More Potent Platelet Inhibition without Increasing Bleeding in Resuscitated Patients

**DOI:** 10.3390/jcm7110442

**Published:** 2018-11-15

**Authors:** Florian Prüller, Lukasz Bis, Oliver Leopold Milke, Friedrich Fruhwald, Sascha Pätzold, Siegfried Altmanninger-Sock, Jolanta M. Siller-Matula, Friederike von Lewinski, Klemens Ablasser, Michael Sacherer, Dirk von Lewinski

**Affiliations:** 1Clinical Institute of Medical and Chemical Laboratory Diagnostics, University Hospital Graz, Graz 8036, Austria; florian.prueller@klinikum-graz.at; 2Department of Cardiology, Medical University of Graz, Graz 8036, Austria; lukasz.bis@stud.medunigraz.at (L.B.); sascha.paetzold@klinikum-graz.at (S.P.); siegfried.altmanninger-sock@klinikum-graz.at (S.A.-S.); friederike.von-lewinski@medunigraz.at (F.v.L.); klemens.ablasser@medunigraz.at (K.A.); 3Department of Anaestiology and Intensive Care, Kardinal Schwarzenberg Hospital, Schwarzach im Pongau 5620, Austria; oliver.milke@ks-klinikum.at (O.L.M.); Michael.sacherer@medunigraz.at (M.S.); 4Department of Cardiology, Medical University of Vienna, Vienna 1090, Austria; friedrich.fruhwald@medunigraz.at (F.F.); jolanta.siller-matula@meduniwien.ac.at (J.M.S.-M.)

**Keywords:** platelet function, resuscitation, cangrelor, light transmission tomography, myocardial infarction, acute coronary syndrome

## Abstract

Dual antiplatelet therapy is the standard of care for patients with myocardial infarction (MI), who have been resuscitated and treated with therapeutic hypothermia (TH). We compare the antiplatelet effect and bleeding risk of intravenous cangrelor to oral P2Y12-inhibitors in patients with MI receiving TH in a prospective comparison of two matched patient cohorts. Twenty-five patients within the CANGRELOR cohort were compared to 17 patients receiving oral P2Y12-inhibitors. CANGRELOR group (NCT03445546) and the ORAL P2Y12 Group (NCT02914795) were registered at clinicaltrials.gov. Platelet function testing was performed using light-transmittance aggregometry and monitored for 4 days. P2Y12-inhibition was stronger in CANGRELOR compared to ORAL P2Y12 (adenosine diphosphate (ADP) (area under the curve (AUC)) 26.0 (5.9–71.6) vs. 160.9 (47.1–193.7)) at day 1. This difference decreased over the following days as more patients were switched from CANGRELOR to oral P2Y12-inhibitor treatment. There was no difference in the effect of aspirin between the two groups. We did not observe significant differences with respect to thrombolysis in myocardial infarction (TIMI) or Bleeding Academic Research Consortium (BARC) classified bleedings, number of blood transfusions or drop in haemoglobin B (Hb) or hematocrit (Hct) over time. Cangrelor treatment is not only feasible and effective in resuscitated patients, but also inhibited platelet function more effectively than orally administered P2Y12-inhibitors without an increased event rate for bleeding.

## 1. Introduction

Dual antiplatelet therapy is the standard treatment for preventing reinfarction and stent thrombosis after coronary stent implantation [1]. According to the guidelines, potent oral P2Y12 inhibitors should be used instead of clopidogrel in acute coronary syndrome [2]. A new P2Y12-receptor inhibitor, cangrelor, has recently become available. In patients suffering from myocardial infarction complicated with resuscitation and having received therapeutic hypothermia, additional challenges must be taken into consideration when determining the course of treatment. Crushing P2Y12 inhibitor tablets is no longer considered a therapeutic limitation as it has not been associated with a reduction in inhibitory effects [3,4,5].

Furthermore, crushing ticagrelor tablets is associated with an even stronger ticagrelor effect than not crushing the tablets [6]. Multiple trials indicate that there is no reduction or delay in platelet inhibition after crushing the tablets, and instead found that there was both an acceleration and increase in the mode of action when crushed. Three other issues remain under debate. First, gastric paresis and reduced gastric and intestinal perfusion due to reduced cardiac output and centralised circulation might reduce or delay effects. Second, hypothermia itself impacts platelet inhibition. Third, resuscitation with consecutive inflammation might directly impact platelet reactivity. Moreover, different impact on platelet calcium homeostasis between the P2Y12-inhibitors might also result in altered pharmacological efficacy. While the first problem can be solved by intravenous administration of the therapy, the second and third problem might impair platelet inhibition independent of the route of administration in this group of patients who are particularly at risk for death and cardiac adverse events. There is limited data for intravenous P2Y12 inhibitors available from cardiogenic shock trials that also includes several resuscitated patients [7,8] and there are no prospective trials regarding the use of intravenous P2Y12 inhibitors in resuscitated patients. We hypothesised that cangrelor is more potent at inhibiting platelet function compared to oral P2Y12 inhibitors at a comparable bleeding risk. We are the first study to present data regarding the use of cangrelor in this vulnerable patient population and compare the effects to the standard treatment via gastric line.

## 2. Materials and Methods

A total of 25 hospital resuscitated patients with the diagnosis of troponin-positive ACS, interventional confirmed coronary heart disease (15 non ST-segment elevation myocardial infarction (NSTEMI) or 10 ST-segment elevation myocardial infarction (STEMI)), and survival until the morning after the index event were consecutively included in this prospective, observational, non-randomised single-centre study. Cangrelor was administered via a bolus in the catheterization lab (30 µg/kg), followed by continuous infusion of 0.75 µg/kg/min as identified and tested in the BRIDGE trial [9]. It was at the interventionists’ discretion to increase the infusion rate of cangrelor to 4 µg/kg/min for 2–4 h, as used in the CHAMPION-PHOENIX trial (9 out of 25 patients) [10]. Cangrelor was administered until gastroparesis was overcome, resulting in a switch towards oral P2Y12 inhibitors in the majority of patients within the four days observational period (CONSORT flow diagram, Appendix A). Combined data of all patients receiving cangrelor is compared to 17 control patients (ORAL P2Y12) of comparable age, gender, and weight who received oral P2Y12 inhibitor (clopidogrel *n* = 4, prasugrel *n* = 5 and ticagrelor *n* = 8) treatment and were derived from a previous trial (NCT02914795) [11]. Demographic data for both groups is listed in Table 1. Patients were enrolled from July 2016 to February 2018. Inclusion criteria were defined as: (1) age >18 years, (2) resuscitation due to acute myocardial infarction, (3) intra-hospital survival until next morning. Clinically significant bleeding leading to discontinuation of anti-platelet therapy before the first platelet function test was used as an exclusion criterion. The study complied with the Declaration of Helsinki, the protocol was approved by the Local Ethics Committee of the Medical University of Graz (No. 28-291 ex 15/16), and registered at clinicaltrials.gov with the ID NCT03445546.

All 25 patients received intravenous aspirin (100–300 mg; average 200 mg) before catheterisation but were not pre-treated pre-clinically with P2Y12 inhibitors due to intubation and ventilation. Eleven patients took aspirin daily (100 mg per day orally) and all patients enrolled were P2Y12 naïve at the time of acute coronary syndrome (ACS). Patients were treated according to the local standard operating procedure, which includes intravascular cooling (Thermogard, Zoll, MA, USA) started immediately after the patient arrives at the intensive care unit (ICU). The target for the cooling procedure was to reduce the core body temperature to 33–36 °C. This temperature was maintained for a total of 24 h, followed by a rewarming period, which could last from 5–20 h with a heating rate of 0.2 °C/h. All patients were intubated endotracheally, mechanically ventilation, and under deep analgosedation, which included administration of morphine and relaxants.

Blood was drawn every morning following the index event between 6 a.m. and 7 a.m. Platelet function testing was performed by light transmittance aggregometry (LTA) on a Chronolog 700 Aggregometer (Chronolog Corp., Havertown, PA, USA). Aspirin reactivity was monitored by inducing platelet aggregation with 2 µg/mL collagen and 0.5 mmol L^−1^ arachidonic acid (AA, Chronolog Corp., Havertown, PA, USA), respectively. P2Y12 inhibition was recorded by stimulation of platelet aggregation with 10 µmol L^−1^ adenosine diphosphate (ADP) (Sigma-Aldrich, Vienna, Austria). An amplitude of ≤20% and/or an area under the curve (AUC) of ≤200 was considered to represent sufficient inhibition. To quantify the overall platelet response, 40 µmol L^−1^ thrombin receptor-activated peptide (TRAP) (Bachem, Weil/Rhein, Germany) was added. Results are displayed using the Aggrolink 8.1.2.2 software package (Chronolog Corp., Havertown, PA, USA).

Data are presented as median (interquartile range). Statistical analyses were performed with IBM SPSS Statistics for Windows Version 22.0. (IBM Corp., Armonk, NY, USA) using the Kruskal–Wallis test, the Wilcoxon signed rank test within groups and the Mann–Whitney U test to test for differences between groups. P-values below 0.05 were considered statistically significant. Power calculation was based on estimated adenosine diphosphate (ADP) area under the curve (AUC) values of 120 and 85 in control and study groups, respectively, with a standard deviation of 40, an alpha of 0.05, and a power of 0.8. Ten percent drop out was calculated, giving a minimum sample size of 24 subjects.

Bleeding was classified according to the thrombolysis in myocardial infarction (TIMI) criteria and the BARC (Bleeding Academic Research Consortium) consensus report [12] and ascertained by retrospective chart review.

## 3. Results

P2Y12-inhibition was significantly greater in cangrelor treated patients compared to control (ADP (% Aggregation) 15.0 (9.0–24.0) (CANGRELOR) vs. 22.5 (17.3–26.5) (ORAL P2Y12 inhibitors) *p* = 0.018 and ADP (AUC) 26.0 (5.9–71.6) (CANGRELOR) vs. 160.9 (53.5–193.7) (ORAL P2Y12 inhibitors) *p* = 0.005) on day 1 (Figure 1A).

This difference vanished within the following days as more patients were switched from the cangrelor group to oral P2Y12 inhibitor treatment (10/24 on day 2, 17/24 on day 3 and 21/24 on day 4) (Figure. 1B). When only looking at patients who received continuous infusion of cangrelor until day 4, the increased inhibitory effect persisted over time (ADP (AUC) 25 (5.1–46.4) (day 1)), 150.3 (79.8 –196.7) (day 2) and 189.7 (97.5–232.5) (day 3), and 120.7 (115.9–120.7) (day 4), respectively).

The effect of aspirin was not different between the two groups (70.3 (31.9–141.1) for CANGRELOR vs. 93.1 (46.7–195.6) for ORAL P2Y12) on day 1 (*p* = 0.595) but revealed significantly higher collagen AUC values (less inhibitory effect) on day 3 (242.3 (176.8–323.7) vs. 256.7 (130–352.2)) and day 4 (260.4 (167.8–356.5) vs. 291.6 (258.5–330.0)) in both groups (CANGRELOR *p* < 0.001 and ORAL P2Y12 *p* < 0.05) (Figure 2A).

Intravenous Aspirin was administered to 5 out of 25 patients through day 4 in the ICU. However, Aspirin-mediated platelet inhibition was not significantly different compared to the 20 patients on oral treatment via gastric tube (collagen AUC values on day 3 (248.3 (168.4–338.7) vs. 242.2 (179.2–326.6) and 4 (238.1 (167.2–343.5) vs. (268.3 (154.8–383.0) (Figure 2B). Arachidonic acid had a robust inhibitory effect in both groups (data not shown), with a trend of decreasing from day 1 (23.6 (6.0–40.9) for CANGRELOR and 31.5 (16.8–73.3) for ORAL P2Y12) to day 4 (36.3 (12.8–96.9) for CANGRELOR and 25.1 (16.1–74.5) for ORAL P2Y12). There was a borderline correlation of cangrelor mediated platelet inhibition with the severity of disease as measured with the SOFA score (r = −0.076; *p* = 0.623) (Figure 3).

We did not observe significant differences between groups in terms of bleeding. As shown in Table 2, we observed 2/25 major bleeding incidents in the cangrelor group (access site bleeding on day 1 in 1 patient and inguinal bleeding due to shearing of a venous side branch after cooling catheter insertion in another patient) and 4/17 major bleeding incidents with oral P2Y12 inhibitors (severe epistaxis on day 1, pulmonary bleeding in the first three days, hematothorax and access site bleeding at the femoral vein and jugular vein, respectively). In all patients with TIMI major bleeding, the thrombocyte count was >125,000. Two of the six patients had significantly elevated INR levels (1.82 and 1.89) due to oral anticoagulation treatment. Both major bleedings in the CANGRELOR group occurred in patients receiving the 4 µg/kg/min infusion for 2 h before decreasing to 0.75 µg/kg/min. TIMI minor bleeding occurred more frequently as haematocrit dropped by more than 12% without observed blood loss in most of the patients (19/25 patients treated with cangrelor and 9/17 patients treated with oral P2Y12 inhibitor). Average values for haemoglobin and haematocrit are listed in Figure 4A,B. There were comparable decreases over time in both groups and the subgroup of patients who received continuous cangrelor infusion for at least three days.

With respect to BARC classifications, we did not observe any BARC 4 or 5 bleeding during the study period. Only 3 occurrences of Type 3 bleedings and 1 occurrence of type 2 bleeding were documented in patients treated with cangrelor, and 4 occurrences of type 3 and 1 occurrence of type 2 bleeding were documented in patients who received oral treatment. No bleeding (BARC 0) occurred in 4/17 and 4/25 patients, respectively. In addition, we identified one patient during the screening period with fatal bleeding who received cangrelor treatment. This patient died only several hours after intervention and before inclusion in the study. The INR of this patient was 4.2, although no treatment with oral anticoagulation was documented in any of the available medical records.

Overall, 1 patient died in the CANGRELOR group within the 4-day observation period, however this death was not associated with a major bleeding. 

The duration of the ICU stay was not different between the groups (10 (5–14) days for CANGRELOR vs. (12 (7–18) days ORAL P2Y12; *p* = 0.326)).

## 4. Discussion

Sufficient platelet inhibition is crucial in a vulnerable cohort of patients, such as those who have undergone resuscitation after MI, are often unstable hemodynamically, and at risk for developing a stent thrombosis. P2Y12 inhibition with more potent drugs (ticagrelor vs. clopidogrel) has proven to lower the stent thrombosis rate in this setting [13]. In this prospective observational study, we show that platelet function inhibition was sufficient and strong in all patients treated with cangrelor. Average values for ADP on day 1 indicated significantly stronger platelet inhibition compared to oral P2Y12 inhibitors. This stronger inhibitory effect could either result from the dosages used or form different pharmacological activities including absorption, distribution, metabolism and excretion. Moreover, differences in platelet calcium homeostasis might also directly impact platelet function, as shown in various models ranging from murine platelets [14] to distinct animal models [15,16]. However, comparative data of oral and intravenous P2Y12-inhibitors on platelet calcium homeostasis is not yet available. Reduced platelet inhibition using clopidogrel as well as newer oral P2Y12 inhibitors is a highly discussed topic in regard to hypothermia [11,17,18,19]. Although the effect was less pronounced compared to CANGRELOR treatment, we still observed therapeutic platelet inhibition following administration of oral drugs via gastric line in all patients in our control group. However, the first platelet function test took place >8 h after the start of treatment and this potentially underestimates delayed pharmacodynamics within the first few hours. In general, cangrelor ensures an immediate antiplatelet effect that starts during the percutaneous intervention (PCI) due to its rapid onset after bolus administration (~2 min). This same onset does not take place following the administration of crushed oral P2Y12 inhibitors after placement of a gastric line in the cath lab or even following PCI while patients are in the ICU, as recently reported for clopidogrel [20], prasugrel [21], and Ticagrelor [22].

Bleeding complications were analysed with retrospective chart analysis. With respect to the TIMI classification, this approach has been shown to detect significantly more bleeding episodes compared to the blinded prespecified documentation in the CHAMPION PHOENIX trial [23]. Using the TIMI classification, we counted a large number of TIMI minor bleeding events due to a decrease in haematocrit of more than 12% without observed blood loss during the study period. This is likely driven by intravenous volume substitution in these patients, as recommended by our local standard operating procedure (SOP). Therefore, using BARC classifications might be more suitable for identifying non-major bleeding events in our cohort. However, both approaches did not indicate significant differences with respect to bleeding between the groups, although bleeding seemed to be more heterogenous in the ORAL P2Y12 group with a higher number of TIMI major bleedings as well as no bleedings, while the vast majority of bleedings in the CANGRELOR group could be classified as TIMI minor bleedings.

Within this study, 10 patients were given the initial higher infusion rate of 4 µg/kg/min for 2–4 h before being switched to a lower infusion rate of 0.75 µg/kg/min. The other 15 patients only received the infusion rate of 0.75 µg/kg/min after the initial bolus. The two major bleeding events in the cangrelor group occurred in patients receiving the higher infusion rate for the first 2–4 h. Although the numbers of patients in this study is certainly too small to make any firm recommendations, it might be advisable to withhold administering the 2–4 h high dose infusion of 4 µg/kg/min in patients with certain bleeding risks. Since we did not observe any suspected or proven stent thrombosis in the cangrelor group patients, the inhibitory effect might be strong enough with the lower dose in these patients. The BRIDGE trial previously revealed that the lower infusion dose of 0.75 µg/kg/min resulted in a potent platelet inhibition of <240 peripheral resistance unit (PRU) in all but one patient (98.8%) [8]. The applicability of lower cangrelor doses might also be supported by a pharmacodynamic study indicating comparable effects of 30 µg/kg bolus and 4 µg/kg/min infusions compared to 15 µg/kg bolus and 2 µg/kg/min infusions in healthy volunteers [24]. As previously reported, 36 out of 12,565 patients treated with cangrelor in the CHAMPION program were accidentally overdosed by more than 20%, but there was no significant increase in the bleeding risk of these overdosed patients. Furthermore, no significant effect of overdosing or correlation between the degree of overdosing and bleeding was observed [25].

Presently, the bleeding risk in ACS patients is lower than in older studies, as most hospitals favour radial over femoral access in these patients, as recommended by the guidelines (ESC STEMI and revascularisation guideline) and supported by large trials [26]. However, in resuscitated patients, femoral access is necessary independent of coronary angiography and PCI if invasive cooling is performed, which seems to be a superior method compared to surface cooling with respect to time spent at the target temperature, and continuous and slow rewarming [27,28]. Therefore, access site bleeding is of increased importance in these patients, which was highlighted by one of our patients who developed a TIMI major bleeding due to a venous lesion after insertion of the cooling catheter.

Another critical time point in the management of platelet inhibition management is the transition to oral treatment. In our study, it was performed by administration of oral medication 30 min before termination of cangrelor infusion. This timeline was used to prevent competitive interaction between the active metabolites of clopidogrel and prasugrel with cangrelor at the level of the P2Y12-receptor as demonstrated by in vitro studies [29,30]. In order to allow enough time for gastric absorption, a necessary 30-min waiting period between administration of the oral medication and termination of the cangrelor infusion was incorporated into the protocol to minimise the potential gap of reduced inhibition during the transition. Recent data suggests that prasugrel could be given even 2 h before discontinuation of cangrelor, whereas clopidogrel-mediated platelet inhibition was significantly impaired under these conditions if tested 2 h after stopping cangrelor infusion [31].

This study also confirms the recently reported impaired inhibitory effect of oral Aspirin in patients who suffered from cardiac arrest outside of the hospital and were treated with PCI and therapeutic hypothermia [11].

## 5. Limitations

The average cooling temperature was slightly higher in the CANGRELOR group since more patients received target temperature management of 36 °C, as this cohort of patients was treated more recently and the target temperature was modified based on data from the TTM trial [32]. Moreover, platelet function tests in the historical ORAL P2Y12 group were only available on working days, resulting in a reduced number of tests.

## 6. Conclusions

In conclusion, we found that cangrelor administration is feasible in resuscitated patients. Platelet inhibition was effective in all patients, but significantly greater with cangrelor treatment compared to oral P2Y12 inhibitors. Since body temperature itself did not correlate with P2Y12 dependent platelet inhibition, differences between the two groups are rather a matter of gastric/intestinal malabsorption, paresis, or different doses of the drugs used.

Aspirin-mediated platelet inhibition decreases within the first days following resuscitation independent of P2Y12 inhibitors used, supporting a previous finding with oral P2Y12 inhibitors only.

## Figures and Tables

**Figure 1 jcm-07-00442-f001:**
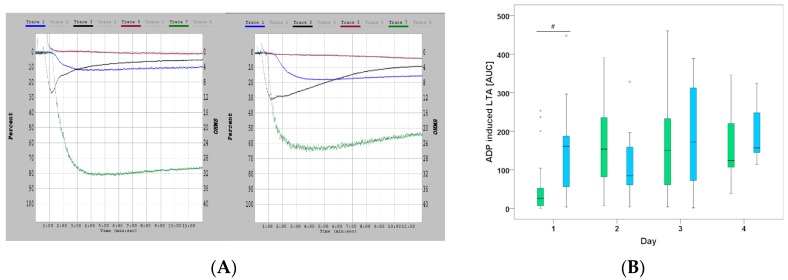
(**A**) Representative original recordings of aggregation curves in a patient on cangrelor treatment (left) and another patient on oral P2Y12-inhibitor treatment (right) Platelet activating compounds: collagen (blue), adenosine diphosphate (ADP) [black], arachidonic acid [red], and thrombin receptor activator peptide (green). (**B**) Median ADP AUC for the first 4 days showing significant difference on day 1. CANGRELOR (green) vs. ORAL-P2Y12 (blue).

**Figure 2 jcm-07-00442-f002:**
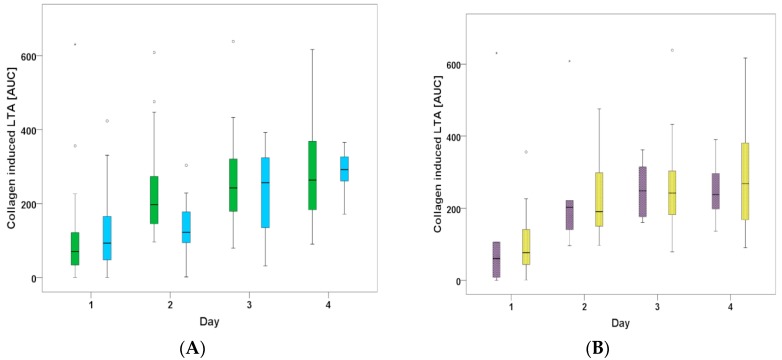
(**A**) Median collagen AUC for the first 4 days. CANGRELOR (green) vs. ORAL-P2Y12 (blue). (**B**) Median collagen AUC for the 25 patients of the CANGRELOR group. Patients with intravenous ASS therapy (black) or oral therapy (gray) after day 1.

**Figure 3 jcm-07-00442-f003:**
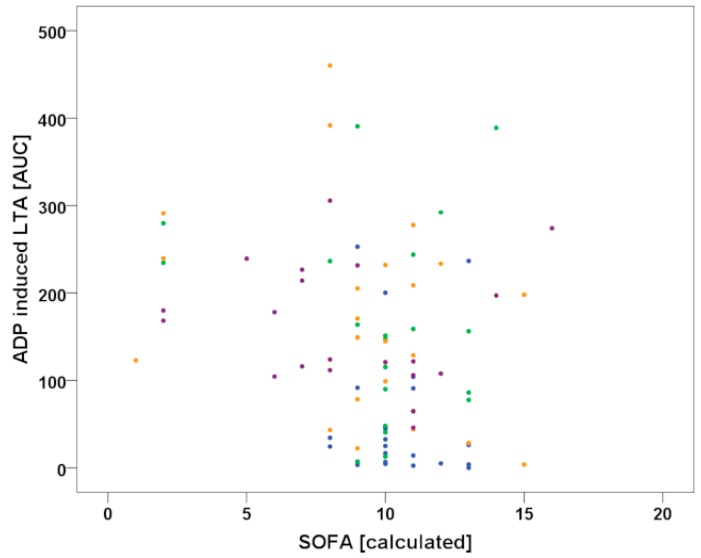
Correlation of SOFA score with ADP-AUC. Data of the 25 patients of the CANGRELOR group is shown. Day 1 (blue), day 2 (green), day 3 (orange), and day 4 (purple) had no differences. Overall data showed r = −0.076; *p* = 0.623).

**Figure 4 jcm-07-00442-f004:**
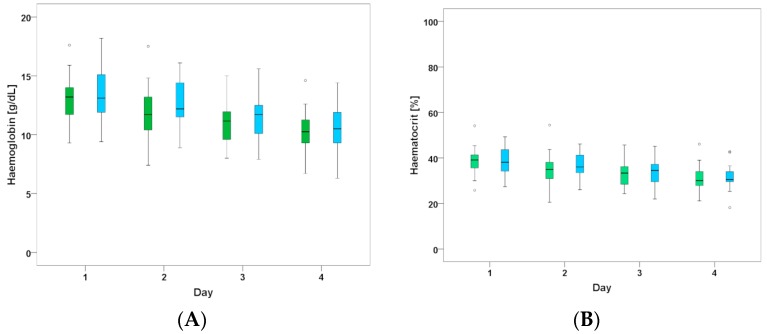
(**A**) Median haemoglobin for the first 4 days after index event. CANGRELOR (green) vs. ORAL-P2Y12 (blue); (**B**) Median haematocrit for the first 4 days after index event. CANGRELOR (green) vs. ORAL-P2Y12 (blue).

**Table 1 jcm-07-00442-t001:** Baseline characteristics, risk factors, and medical history data of the CANGRELOR group (left) and the ORAL P2Y12 group right. No significant differences could be detected between the groups.

*n* (%) or Median (IQR)	Study Group Cangrelor (*n* = 25)	Control-Oral P2Y12 Inhibitor (*n* = 17)	*p*-Value
Characteristics			
Age (years)	61.3 (53.8–71.7)	65.2 (49.3–76.5)	0.750
Malesex	21 (84.0)	14 (82.4)	0.892
STEMI	15 (60.0)	9 (52.9)	0.446
Coronary angiography			
1-vessel disease	13 (52.0)	18 (47.1)	
2-vessel disease	6 (24.0)	6 (35.3)	0.956
3-vessel disease	6 (24.0)	3 (17.6)	
Laboratory parameters			
Platelet count (G/1)	228.0 (189.0–302.0)	185.5 (170.5–214.8)	0.102
aPTT (s)	51.0 (31.8–160)	61.2 (33.9–160)	0.101
INR (1)	1.2 (1.1–1.3)	1.2 (1.1–1.6)	0.720
Hb (g/l)	13.8 (12.7–15.6)	13.9 (12.3–14.9)	0.953
Hct (%)	40.3 (37.7–45.1)	40.9 (35.6–43.4)	0.934
Risk profile/medical history			
Body mass index (kg/m^2^)	26.2 (24.6–30.3)	29.4 (26.2–30.9)	0.492
Alcoholic disease	4 (16.0)	2 (11.8)	0.810
Nicotine abuse	12 (48.0)	7 (41.2)	0.738
Arterial hypertension	13 (52.0)	9 (52.9)	0.130
Diabetes mellitus	5 (20.0)	3 (17.6)	0.929
Hyperlipidemia	11 (44.0)	8 (47.1)	0.443
Peripheral vascular disease	4 (16.0)	3 (17.6)	0.689
Cerebral ischemia	4 (16.0)	1 (5.9)	0.206
Myocardial infarction	3 (12.0)	2 (11.8)	0.980
Coronary angiography	4 (16.0)	5 (29.4)	0.251

IQR, Inter-Quartil-Range; STEMI, ST-segment elevation myocardial infarction; aPTT, activated partial thromboplastin time; INR, international normalised ratio; Hb, Hemoglobin; Hct, Hematocrit.

**Table 2 jcm-07-00442-t002:** Bleeding events in accordance to the TIMI classification (upper part) or the BARC classification (lower part) for the CANGRELOR group (left) and the ORAL P2Y12 group (right).

Bleeding: *n* (%)	Cangrelor	Oral P2Y12
TIMI		
major	2 (8%)	4 (24%)
minor	19 (76%)	9 (53%)
BARC		
4/5	0 (0%)	0 (0%)
3	3 (12%)	4 (24%)
2	1 (4%)	1 (6%)

TIMI, thrombolysis in myocardial infarction; BARC, Bleeding Academic Research Consortium.

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
