# Peer review of "Cangrelor Induces More Potent Platelet Inhibition without Increasing Bleeding in Resuscitated Patients"

_jcm, 2018, doi:10.3390/jcm7110442_

Round 1

Reviewer 1 Report

In this manuscript, the authors compared the antiplatelet effect and bleeding risk of intravenous cangrelor and oral P2Y12 inhibitors. 25 patients were treated with cangrelor and 17 patients received different oral inhibitors. They found that cangrelor inhibited platelet function more efficiency without increasing the bleeding risk. Below are some concerns.

1. The pixel of figures is very low. It is difficult to get the information from the figures.

2. In addition to showing AUC. The representative aggregation curve should also be showed.

3. The authors did not discuss why cangrelor have better effect in inhibiting platelet aggregation compared to other P2Y12 inhibitors. P2Y12 is critical for the initial step of platelet aggregation, and Ca2+ is important for this step (see the literatures, PMID: 24520961; PMID: 26800051; PMID: 29146750). More discussion about this point is important.

Author Response

Thank you very much for your helpful considerations and advice. We appologize for the quality of the original figures you got. We re-formated the figures and do hope that they have better resolution as well as a clearer layout.

In addition to Figure 1A we added representative tracings of the aggregation curves. This advice clearly improves information content for the reader.

We are pleased to add Information and discussion on Calcium-related mechanisms both in the introduction and the discussion and to include the suggested literature.

Reviewer 2 Report

The manuscript by Prüller et al describes the use of cangrelor, a parenteral P2Y12-based antiplatelet, in patients with myocardial infarction, who have been resuscitated and treated with therapeutic hypothermia and compares its effects (platelet inhibition and risk of bleeding) to the standard treatment, P2Y12-based oral antiplatelets, at the authors’ host institution in prospective, observational, non-randomized single-center study. Generally, the manuscript is written in good language. The study is very interesting for clinicians in the field of thromboembolic diseases. The authors obtained all required approvals for the study, adequately presented all relevant experimental and statistical analyses which are in-line with the title and support the conclusions presented at the end. The authors are encouraged to improve the quality of the figures and tables and to provide a list of abbreviations.

Author Response

Thank you very much for your helpful considerations and advice. We appologize for the quality of the original figures you got. We re-formated the figures and do hope that they have better resolution as well as a clearer layout.

A list of abbreviations was added to the manuscript.

Moreover, a new Figure 1B with original tracings was added and the discussion was extended to calcium dependent mechansims.

Round 2

Reviewer 1 Report

The authors addressed my previous concerns. I have no further questions.